# COVID-19 Pandemic and Eating Disorders among University Students

**DOI:** 10.3390/nu13124294

**Published:** 2021-11-28

**Authors:** Marie-Pierre Tavolacci, Joel Ladner, Pierre Dechelotte

**Affiliations:** 1Clinical Investigation Center 1404, CHU Rouen, U 1073, Normandie University, UNIROUEN, F 76000 Rouen, France; 2Department of Epidemiology and Health Promotion, CHU Rouen, U 1073, Normandie University, UNIROUEN, F 76000 Rouen, France; joel.ladner@chu-rouen.fr; 3Department of Nutrition, CHU Rouen, U 1073, Normandie University, UNIROUEN, F 76000 Rouen, France; pierre.dechelotte@chu-rouen.fr

**Keywords:** eating disorder, COVID-19, university students, food security

## Abstract

An online cross-sectional study was conducted in May 2021 to identify factors, such as changes in food choices, lifestyle, risk and protective behavior, mental health, and social demographics, on eating disorders (ED) among students of a French university. Students were invited to fill out an online questionnaire. ED were identified using the French version of the five-item “Sick, Control, One stone, Fat, Food” (SCOFF) questionnaire. The Expali™-validated algorithmic tool, combining SCOFF and body mass index, was used to screen EDs into four diagnostic categories: bulimic ED, hyperphagic ED, restrictive ED and other ED. A total of 3508 students filled the online questionnaire, 67.3% female, mean age 20.7 years (SD = 2.3). The prevalence of ED was 51.6% in women and 31.9% in men (*p* < 0.0001). Lower food security scores were associated with a higher risk for all ED categories. Depression and academic stress due to COVID-19 were associated with ED regardless of category. Regarding health behaviors, a high adherence to the National nutrition recommendation was a protective factor for the risk of bulimic ED, hyperphagic ED and restrictive ED. A lower frequency of moderate and vigorous physical activity was associated with a higher risk of hyperphagic ED. Our study has shown a high screening of ED among the students of a French university fourteen months after the beginning of the COVID-19 pandemic. By disrupting academic learning, jobs and social life, the COVID-19 pandemic could have exacerbated existing ED or contributed to the onset of new ED.

## 1. Introduction

University students may constitute a particularly vulnerable population for mental health problems with the transition to adulthood, frequent economic difficulties and academic burden [1]. University years coincide with the typical age of onset of eating disorders (ED) with a significant concern among university students, worsened by academic stress [2]. To this context was suddenly added the Coronavirus Disease 2019 (COVID-19) pandemic on 11 March 2020 which led to multiple lockdowns in France and curfews [3]. University students have been affected by the lockdown periods with disruptions in teaching with the switch of online learning to continue teaching, leading to worries about adapting to new methods [4]. The implementation of public health measures (including gym closures and prohibited group sports) has caused a decrease in physical activity [5] and a modification of eating habits among university students [6,7]. A recent review of the literature concluded that quarantine measures and the pandemic COVID-19 could have negative psychological effects, including, stress, anxiety, and depression [8,9], higher among university students than non-students [10]. There are a variety of stressors that contribute to increased levels of anxiety, stress, and depressive thoughts as students live through the COVID-19 pandemic [11]. In addition, less socialization, as well as students’ living conditions, may also contribute to lockdown stress [12]. According to Wang et al. the biggest contributor was stress associated with academics, followed by general uncertainty regarding the pandemic, health concerns, and concerns related to finances (loss of job) [13]. Salazar-Fernandez et al. described the role of emotional distress as a key mechanism to explain coping behaviors, such as eating comfort food, adopted as a consequence of the COVID-19 related stressors [14]. University students may have modified or stopped their paid activity [15] which could have led to food insecurity [16]. In the United States, studies showed that university students with food insecurity were more likely to screen positive for an ED [17], especially during the COVID-19 pandemic [18]. The COVID-19 pandemic has also led to anxiety and depression symptoms among university students [19,20] which are well known to be associated with ED [21,22]. The Lin study [23] highlights how the COVID-19 pandemic has hampered the ability to promptly identify and treat young people with eating disorders. Recently we highlighted a sharp increase in ED in 2021 compared to the previous 10 years [24]. However, few studies have evaluated the risk of onset or worsening of ED and associated factors among the student population [25,26].

This study aimed to investigate the associated factors, such as changes in food choices, lifestyle, risk and protective behavior, mental health, and social demographics, on four ED categories (bulimic ED, hyperphagic ED, restrictive ED and other ED) among students of a French university during the COVID-19 pandemic.

## 2. Methods

The recruitment strategy of university students followed a convenience sampling method. In May 2021, university-wide email distribution lists were used to invite students to participate in the study on the impact of the COVID-19 pandemic. If students were interested in participating, they were asked to follow a link to the survey website. This online anonymous cross-sectional observational study was approved by the local Institutional Review Board (E2020-22).

Students were eligible for inclusion if they were currently enrolled at a higher education institution, aged 18 years or more, and accepted to answer the study questionnaire. No duplicates were found (checking on age, gender, cursus, and year of study). Students aged over 30 years were secondarily excluded. There was no missing data because filling in was mandatory.

The study encompasses the pandemic period one month before May 2021 and for eating habits, food security, and physical activity, encompasses also the pre-pandemic period: i.e., the month before the COVID-19 measures (before March 2020). The same questions were asked indicating before the pandemic COVID 19 (before March 2020) and the month before. 

### 2.1. Socio-Demographic Characteristics

Data were collected on age, gender; type of academic course: law and economics, social sciences, health sciences, and technology; year of the academic course, further categorized in year 1, years 2 and 3, years 4 and 5, and year 6 and more.

### 2.2. Eating Habits

The PNNS-GS2 (Programme National Nutrition Santé—Guidelines Score 2) is a predefined food-based dietary index designed to reflect healthy food groups [27]. The PNNS-GS2 recommends six food groups: fruit and vegetables, nuts, legumes, whole-grain food, milk and dairy products, fish, and seafood. A score reflecting adherence to the 2017 French nutritional guidelines was used with weighting according to the level of evidence of the association between food groups and health: a score of 3 for fruit and vegetables, a score of 2 for whole-grain food and fish and seafood, and a score of 1 for nuts, legumes and milk and dairy products [28]. PNNS-GS2 components and scoring (0 to 14) are presented in annexe 1.

### 2.3. Food Security

Food security was assessed using the six-item Food Security Scale [29] with a score of 0–1 indicating high food security, a score of 2–4 indicating low food security, and a score of 5–6 indicating very low food security.

### 2.4. Physical Activity

Students reported their frequency of moderate physical activity, including cycling or walking for at least 30 min, and vigorous physical activity, including lifting heavy weights, running, aerobics, or fast cycling for at least 30 min [30]. Frequency was categorized as (almost) never, less than once a week, once a week (categorized as occasional), more than once a week and (almost) daily (categorized as regular) [5].

### 2.5. Eating Disorders

Students filled in the French version of the five-item “Sick, Control, One stone, Fat, Food” (SCOFF) questionnaire. A diagnostic threshold was fixed at two positive responses, with a sensitivity of 0.88 and a specificity of 0.93, using interviews as a diagnostic reference; therefore, data obtained with SCOFF gave a proxy of actual ED The Expali™-validated algorithmic tool, combining SCOFF and body mass index, was used to screen EDs into four diagnostic categories: bulimic ED, hyperphagic ED, restrictive ED and other ED (purging disorder, Night Eating Syndrome and any other ED) [31].

### 2.6. Mental Health

Depression was assessed using the eight items of the CESD-8 (Center for Epidemiologic Studies-Depression) scale, which has shown adequate psychometric properties (a Cronbach alpha of 0.82) [32]. The response values were scored on a 4-point Likert scale (range 0 to 3) and CESD-8 on a scale from 0 to 24, with higher scores indicating a higher frequency of depressive complaints.

Academic stress was assessed using a Likert scale (0 totally disagree to 4 totally agree) according to increased academic workload, the concern of not being able to validate the academic year, stress with changes in teaching methods, and difficulty in keeping up e-learning courses (insufficient equipment, weariness), then academic stress was scored from 0 to 16. The academic stress and academic satisfaction scale have sufficiently high internal consistency (Cronbach’s alphas are greater than 0.6 [33].

Students’ concern about COVID-19 was assessed on a Likert scale from 0 to 10 based on the following items: worry about becoming severely ill from a COVID-19 infection; worry about a relative becoming severely ill from a COVID-19 infection.

Students were asked if they had visited a healthcare professional for support during the year 2021 and if yes, which category of healthcare professional: general practitioner, psychologist, psychiatrist, and nutritionist.

## 3. Statistical Analysis

Qualitative variables were summarized by percentage and compared using the Chi^2^ test and continuous variables by mean with standard deviation (SD) and compared using the Student Test. Cross-sectional associations were estimated via multivariable polytomous logistic regression (no ED = reference category) providing adjusted ORs and 95% CIs. The principal outcome (dependent) variable was the 4-category ED measure (restrictive, bulimic, hyperphagic and other ED). Variables with *p* value < 0.20 were included in the logistic regression. A *p* value below 0.05 was considered to be significant. The analysis was conducted using XLSTAT by Addinsoft, Paris, France 1 March 2020.

## 4. Results

A total of 3508 students (response rate of 12%) filled the online questionnaire; 67.3% female, mean age 20.7 years (SD = 2.3). Among this sample of students, 10.7% were underweight, 13.5% were overweight and 5.4% were obese. The characteristics of the students are presented in Table 1.

### 4.1. Eating Habits

PNNS-G2 score was lower during the COVID-19 pandemic in students with ED (4.4 SD = 2.4) than in students with no ED (4.9 SD = 2.3). The score by category of ED was displayed in Table 1. PNNS-G2 score decreased for bulimic ED, hyperphagic ED, and restrictive ED between pre and pandemic periods (Figure 1).

### 4.2. Food Security

During the COVID-19 pandemic, 11.2% of students with no ED and 26.3% of students with ED had low and very low food security (*p* < 0.0001) (Table 1), with an increase in food insecurity between pre and pandemic COVID-19 period (Figure 2).

### 4.3. Physical Activity

Moderate and vigorous PA was lower in students with ED than in students without ED and the lowest PA among students with hyperphagic ED (Table 1). The frequency of moderate physical activity and vigorous physical activity decreased in students with ED and in students with no ED (Figure 3A,B).

### 4.4. Eating Disorders

The screening of ED was 51.6% in women and 31.9% in men (*p* < 0.0001); half of ED were bulimic ED. The distribution by category of ED according to gender is presented in Figure 4.

### 4.5. Mental Health

CESD-8 score was higher among students with ED compared to students with no ED (13.4 SD = 5.6 and 12.0 SD = 3.3; *p* < 0.0001) as academic stress score (9.0 SD = 5.4 and 10.2 SD = 3.8; *p* < 0.0001). Scores for the items “worry about becoming severely ill” and “worry about a relative becoming severely ill” were higher among students with ED compared to students with no ED (3.8 SD = 3.3 and 3.4 SD = 31; *p* < 0.001) and (8.1 SD = 2.5 and 7.5 SD = 2.7; *p* < 0.0001), respectively. Students with ED consulted more frequently a healthcare professional (21.7%) than students with no ED (11.3%) (*p* < 0.0001). These results are detailed by ED category in Table 1. Psychologists were the most frequently consulted healthcare professionals. There was no difference in the category of healthcare professionals consulted according to the presence or absence of an ED (Figure 5). 

Results after multivariate analysis were displayed in Table 2. The significant associated factors (*p* < 0.05) were the following. Female gender, social humanities and law/economic curricula were risk factors for all ED categories. Students in years 1 and 2 had a higher risk of bulimic and restrictive ED than students in years 3 or more. Lower food security scores were associated with a higher risk for all ED categories. Depression, academic stress and consultation of a healthcare professional were associated with ED regardless of category. Regarding health behaviors, a high PNNS G2 score was a protective factor for the risk of bulimic, hyperphagic and restrictive ED. A low frequency of moderate physical activity was associated with an increased risk of bulimic ED and hyperphagic ED and a low frequency of vigorous physical activity was associated with an increased risk of hyperphagic ED. 

## 5. Discussion

This cross-sectional study provides the screening of ED fourteen months after the beginning of the COVID-19 pandemic declared on 11 March 2020 [3] and shows ED affected one in two female students and one in three male students in May 2021. These results highlight an increase in ED, and support the results of a recent study conducted among students of the same university in the past decade years which showed that one in three women and one in seven men had an ED [24]. Lin et al. [23] demonstrated the increasing volumes of inpatient and outpatient young adults with eating disorders since the COVID-19 pandemic began. Bulimia remained the most prevalent ED as before the COVID-19 pandemic [24]. In our study, students with bulimia were especially worried about becoming severely ill. Among a population of patients, more severe COVID-19-related post-traumatic symptomatology was reported in patients with bulimia than in patients with anorexia and hyperphagia [34]. Severe anxiety was also associated with an increase in hunger, emotional over-eating, and a decrease in enjoyment of food. Our study allowed to identify depression and added stress related to COVID-19 (academic disruption and fear of infection) associated with a risk of ED that may also explain the increased prevalence of ED [16,18]. We found an increase in the food insecurity and an association with each category of ED, it was already found to be associated with increased binge-eating disorders in the general population before the COVID-19 pandemic [35]. One possible explanation is that individuals who lack adequate resources to regularly purchase enough food to meet their nutritional needs undergo cycles of food restriction. These bouts of restriction may increase the risk of binge eating via food cravings or the biological effects of starvation. Another, possibly complementary, explanation is that economic strain creates stress, which in turn may promote binge eating [36]. At the beginning of the COVID-19 pandemic, patients with anorexia nervosa had increased restrictions, and feared being able to find foods consistent with their meal plan while patients with bulimia nervosa or with a binge eating disorder had increased binges [37].

Our study shows that students with ED consulted a healthcare professional twice more than students with no ED during the COVID-19 pandemic, apparently not foregoing care. Direct care was often replaced by tele-medicine in order to continue providing care while minimizing the risk of transmission of COVID-19 [38]. However, by disrupting typical modes of service delivery such as in-person office visits with a health-care provider, the COVID-19 pandemic may have exacerbated the already pervasive problem of unmet treatment needs among individuals with an eating disorder [39]. Tele-medicine education should be more developed in the curriculum of healthcare students to be effective beyond the pandemic COVID-19 [40].

Regarding health behavior, a decrease in physical activity was observed among university students during the pandemic with a greater risk of absence of physical activity among university students with hyperphagic ED and bulimic ED. In clinical populations, patients with anorexia are known for physical hyperactivity [41] and patients with bulimic/hyperphagic ED for sedentarity [42]. Fernandez-Aranda et al. reported changes in eating habits in patients with ED linked to an increase in restrictive diets due to concerns about weight and shape, and phases of binge eating seem to be: greater sedentary lifestyle, restrictions on outdoor activities, reduction of physical exercise, alterations in the sleep-wake rhythm and fear of contagion [43].

## 6. Limitation

Caution is advised when generalizing these findings, for the following reasons: first, investigating health behaviors may lead to errors in self-reporting, particularly shifts in how body weight is perceived, reflecting cognitive distortions that could increase the risk for disordered eating in some individuals [44]; second, this was a convenience sample, and voluntary participation could have led to representativeness and self-selection bias as our sample had more women and healthcare students, third the study was cross sectional and therefore, does not allow causal interpretation between risk factors and ED. The study being anonymous and by self-questionnaire limits the bias of desirability.

## 7. Conclusions

Protecting the mental health of students is a public health issue that appears even more critical in the context of a pandemic. It also appears important that students can maintain physical activity social ties, and financial resources. This is also an opportune time to rethink prevention, early identification, in the post-COVID-19 era and in the future, should another pandemic hit. Students with self-stigma could endorse e-therapy which was effective in reducing eating disorder symptoms and comorbid depressive or anxiety symptoms.

## Figures and Tables

**Figure 1 nutrients-13-04294-f001:**
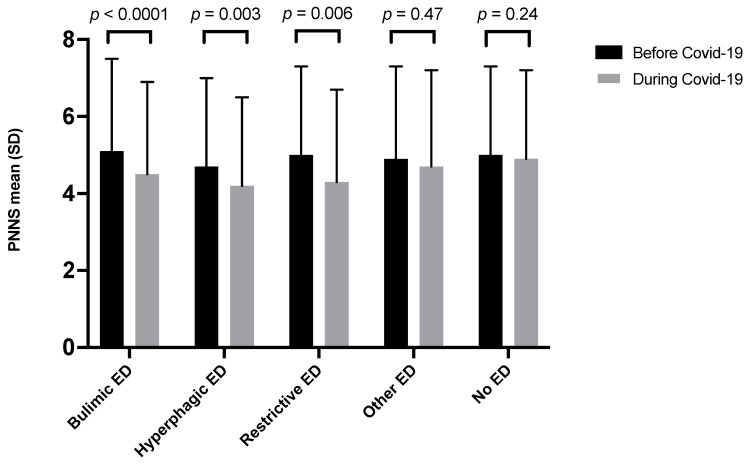
PNNS-G2 score before and during the COVID-19 pandemic according to the category of ED N = 3508.

**Figure 2 nutrients-13-04294-f002:**
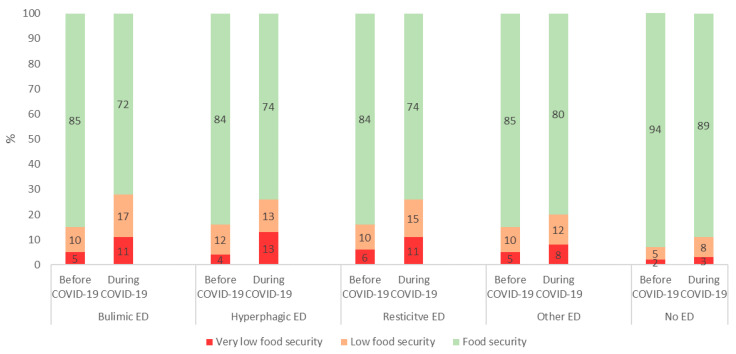
Food insecurity before and during the COVID-19 pandemic according to the category of ED N = 3508. (*p* < 0.05 for each category).

**Figure 3 nutrients-13-04294-f003:**
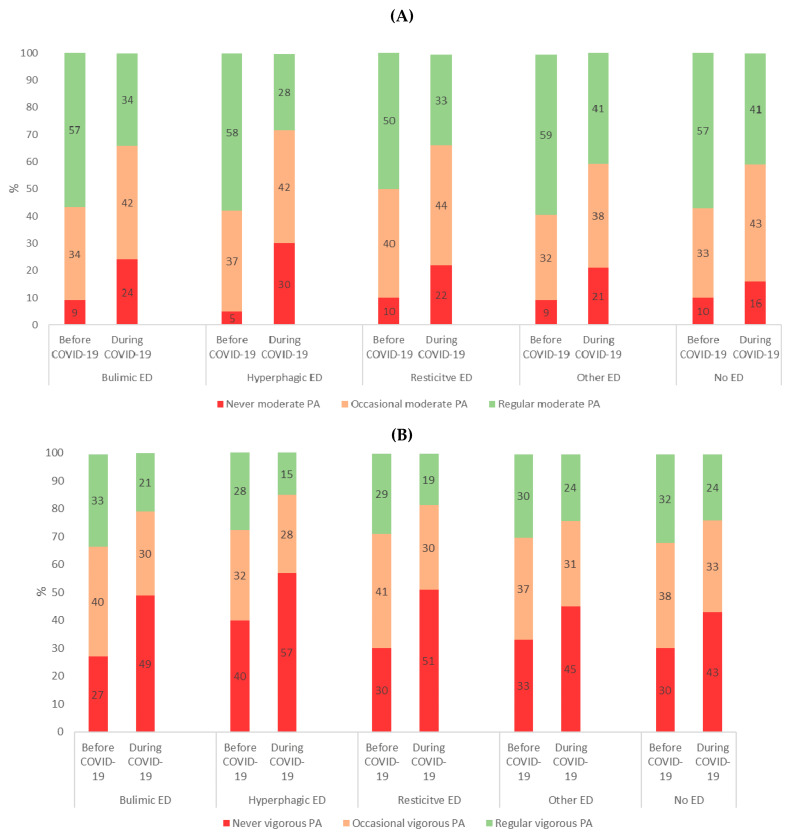
(**A**): Moderate physical activity and, (**B**): vigorous physical activity before and during COVID-19 according to ED category N = 3508. (*p* < 0.05 for each category).

**Figure 4 nutrients-13-04294-f004:**
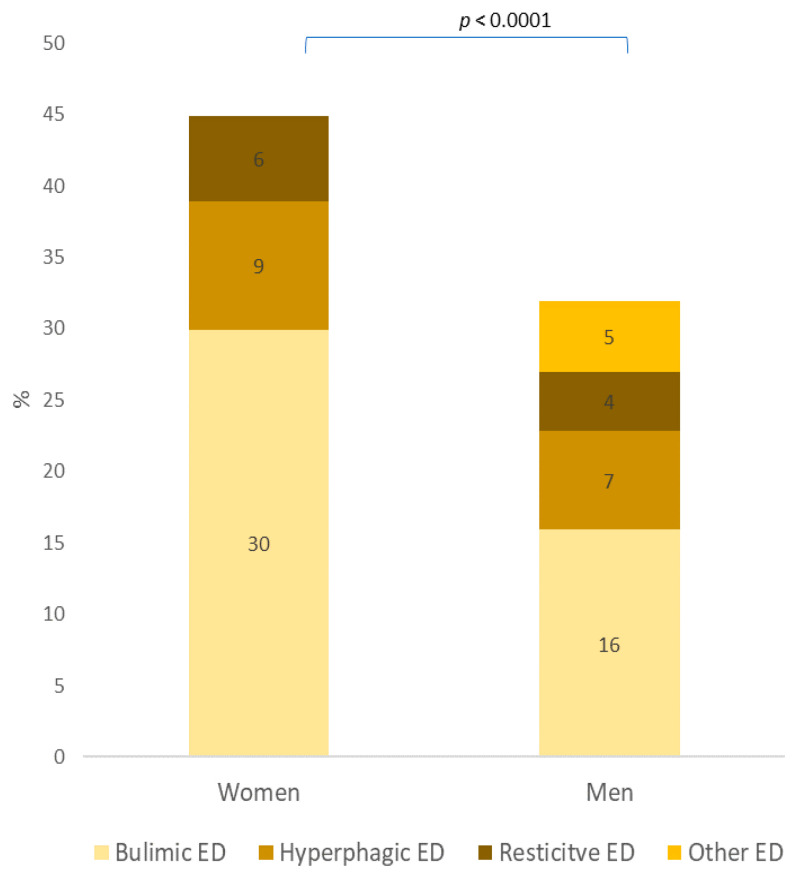
Category of eating disorder according to gender (N = 3508).

**Figure 5 nutrients-13-04294-f005:**
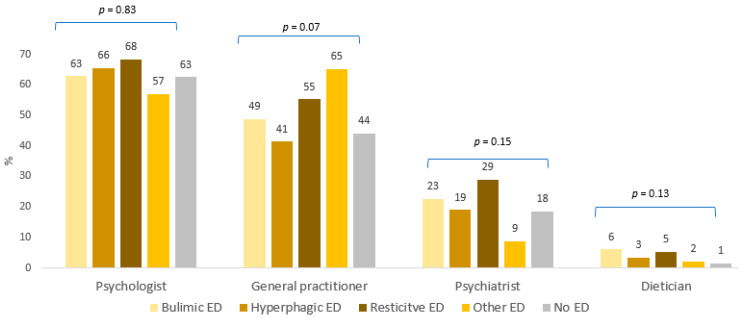
Category of healthcare professional consulted *N* = 566.

**Table 1 nutrients-13-04294-t001:** Characteristics of university students according to the category of eating disorder (univariate analysis) *N* = 3508.

	Bulimic ED *n* = 924	Hyperphagic ED *n* = 297	Restrictive ED *n* = 195	Other ED *n* = 218	No ED *n* = 1874	*p*	Total *N* = 3508
Women	84.5	79.1	80.5	79.8	67.3	<0.0001	74.4
Curriculum						<0.0001	
Health sciences	29.6	26.3	24.1	28.4	34.2		31.4
Technology	22.8	21.5	20.0	20.7	27.2		24.7
Social sciences	28.2	32.3	32.3	32.1	22.9		26.2
Law and economy	19.3	19.9	23.6	18.8	23.6		17.6
Year of study						0.005	
1	39.7	34.3	42.0	36.7	34.2		36.3
2	25.9	27.9	31.3	24.3	25.6		26.1
3	17.3	18.6	14.9	17.4	17.6		17.4
4 and more	17.1	18.6	11.8	21.6	22.6		20.2
PNNS-G2	4.48 (2.39)	4.18 (2.32)	4.32 (2.42)	4.69 (2.54)	4.91 (2.34)	<0.0001	4.69 (2.38)
Moderate physical activity						<0.0001	
Occasional	41.8	41.8	44.1	38.1	42.9		42.2
Regular	34.1	27.8	33.3	41.3	41.0		37.7
Vigorous physical activity						0.001	
Occasional	29.4	28.0	30.3	30.8	32.8		31.2
Regular	21.0	15.1	18.5	23.8	23.7		22.0
Food security						<0.0001	
Low	16.6	13.5	15.4	12.4	7.8		11.3
Very low	11.4	12.5	10.3	8.3	3.4		7.0
Depression (score/24)	13.7 (5.7)	13.4 (.4)	13.8 (5.5)	11.9 (5.6)	9.0 (5.4)	<0.0001	11.1 (5.9)
Academic stress (score/16)	12.1 (3.4)	12.2 (3.1)	10.2 (3.8)	11.4 (3.6)	10.2 (3.8)	<0.0001	11.0 (3.7)
Worry about becoming severely ill (%)	3.7 (3.3)	4.2 (3.4)	3.8 (3.4)	4.0 (3.2)	3.4 (3.1)	0.001	3.6 (3.2)
Worry about a relative becoming severely ill (%)	8.0 (2.5)	8.0 (2.6)	8.3 (2.5)	8.3 (2.3)	7.5 (2.7)	<0.0001	7.8 (2.6)
Consultation of a health professional (%)	23.0	19.5	19.5	21.1	11.2	<0.0001	16.1

*p: p value*.

**Table 2 nutrients-13-04294-t002:** Factors associated with the category of eating disorder (multivariate analysis) *N* = 3508. Adjusted on year of study, gender and curriculum.

	Bulimic ED *n* = 924	*p*	Hyperphagic ED *n* = 297	*p*	Restrictive ED *n* = 195	*p*	Other ED *n* = 218	*p*
Women	2.6 (2.1–3.2)	<0.0001	1.7 (1.3–2.4)	<0.0001	1.9 (1.3–2.8)	0.001	1.8 (1.3–2.6)	0.001
Curriculum								
Health sciences	Ref		Ref		Ref		Ref	
Technology	1.2 (0.9–1.5)	0.15	1.1 (0.8–1.6)	0.15	1.2 (0.7–1.8)	0.48	1.0 (0.7–1.6)	0.80
Social sciences	1.4 (1.1–1.7)	0.005	1.7 (1.3–2.4)	0.005	1.9 (1.2–2.8)	0.002	1.6 (1.1–2.4)	0.007
Law and economy	1.5 (1.1–1.9)	0.002	1.4 (1.1–1.8)	0.002	2.1 (1.4–3.3)	0.001	1.5 (0.9–2.2)	0.06
Year of study								
1	1.5 (1.5–1.9)	0.0001	1.2 (0.8–1.7)	0.36	2.3 (1.4–3.7)	0.001	1.1 (0.7–1.6)	0.62
2	1.3 (1.0–1.6)	0.03	1.2 (0.9–1.8)	0.24	2.2 (1.3–3.7)	0.002	0.9 (0.6–1.4)	0.83
3	1.2 (0.9–1.7)	0.13	1.2 (0.8–1.8)	0.37	1.5 (0.8–2.6)	0.16	1.0 (0.6–1.5)	0.93
4 and more	Ref		Ref		Ref		Ref	
Food security								
Yes	Ref		Ref		Ref		Ref	
Low	2.71 (2.11–3.48)	<0.0001	2.08 (1.42–3.04)	<0.001	2.32 (1.50–3.57)	<0.001	1.80 (1.15–2.80)	0.009
Very low	4.07 (2.93–5.66)	<0.0001	4.21 (2.73–6.49)	0.0001	3.43 (2.00–5.86)	<0.0001	2.61 (1.51–4.52)	0.001
Depression	1.15 (1.13–1.17)	<0.0001	1.14 (1.11–1.16)	<0.0001	1.15 (1.12–1.18)	<0.0001	1.09 (1.06–1.12)	<0.0001
Academic stress	1.15 (1.12–1.18)	<0.0001	1.15 (1.12–1.18)	<0.0001	1.14 (1.09–1.20)	<0.0001	1.09 (1.05–1.14)	<0.0001
Worry about becoming severely ill	1.00 (0.98–1.03)	0.87	1.05 (1.02–1.09)	0.005	1.02 (0.97–1.06)	0.48	1.03 (0.99–1.08)	0.13
Worry about a relative becoming severely ill	1.06 (1.02–1.09)	0.001	1.06 (1.01–1.11)	0.001	1.10 (1.03–1.17)	0.002	1.14 (1.05–1.18)	0.001
Consultation of a health professional	2.27 (1.83–2.82)	<0.001	1.83 (1.32–2.53)	<0.001	1.90 (1.29–2.79)	0.001	1.99 (1.39–2.85)	0.0001
Health behavior								
PNNS-G2	0.93 (0.90–0.97)	<0.001	0.88 (0.84–0.94)	<0.0001	0.91 (0.86–0.98)	0.007	0.97 (0.91–1.03)	0.35
Moderate physical activity								
Never	Ref						Ref	
Occasional	1.11 (0.92–1.33)	0.27	1.35 (1.00–1.82)	0.05	1.64 (0.83–1.64)	0.38	1.24 (0.82–1.79)	0.27
Regular	1.70 (1.36–2.13)	<0.0001	2.64 (1.89–3.67)	<0.0001	1.51 (1.00–2.28)	0.05	0.83 (0.61–1.15)	0.32
Vigorous physical activity								
Never	Ref		Ref		Ref		Ref	
Occasional	1.01 (0.80–1.26)	0.94	1.34 (0.91–1.97)	0.13	1.20 (0.78–1.86)	0.17	0.93 (0.64–1.37)	0.73
Regular	1.15 (0.93–1.41)	0.20	1.87 (1.32–2.66)	<0.0001	1.32 (0.89–1.98)	0.41	0.94 (0.66–1.35)	0.74

no ED = reference category.

## Data Availability

Data are available on request.

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
