# Peer review of "COVID-19 Pandemic and Eating Disorders among University Students"

_nutrients, 2021, doi:10.3390/nu13124294_

Round 1
Reviewer 1 Report
The authors address an interesting subject within the current period. Eating disorders are expected to increase in a time where everyday life and habits are supressed by the pandemic.
The methodological design does not allow though for certain conclusions to be drawn.
Line 65. The authors refer to influence, but the cross sectional design does not allow for such a hypothesis to be examined.
Line 72. How did the authors ensure that the participants answered the questionnaire only once? The authors should provide the procedure under which they were identified and verified for non duplicates. Could anyone following the suggested link fill the questionnaire without providing credentials?
Line 97. The authors should justify the way they measured physical activity with these questions and how they decided the categories they refer to on line 101.
Line 115. Why did the authors measure academic stress with these questions? How was the reliabiblity of the academic stress measurement assessed?
Line121. The authors should justify the use of these questions regarding COVID 19 and not the use of some other validated questionnaire, since many have been developed (e.g. CAS, OCS, CRBS).
Statistical analysis
The statistical analysis presented is generic. The authors should provide in further detail what models they used and under which criteria did reach inference. Were multiple comparison tests applied?
In the last paragraph of the results section (lines 169-178) there is no indication of the p- values that lead to these conclusions.
Table 1. The authors should provide more information on the statistical tests that were applied and led to these conclusions. What hypotheses do the p-values refer to? Multiple comparisons should be carried out to shed light to several estimations that appear on the table.
Table2. Further annotation is needed to help the reader understand what hypotheses are examined by the p-values presented.
Discussion section.
The discussion should emphasize more on the research findings as the presentation of the literarure findings dominate the section.
All figures except figure 4 should contain more statistical information as the conclusions based on the depicted information can be mistaken.
Author Response
The authors address an interesting subject within the current period. Eating disorders are expected to increase in a time where everyday life and habits are supressed by the pandemic.
We thank the reviewer
The methodological design does not allow though for certain conclusions to be drawn.
Line 65. The authors refer to influence, but the cross sectional design does not allow for such a hypothesis to be examined.
The sentence has been modified “This study aimed to investigate the assoiated factors »
Line 72. How did the authors ensure that the participants answered the questionnaire only once? The authors should provide the procedure under which they were identified and verified for non duplicates. Could anyone following the suggested link fill the questionnaire without providing credentials?
To ensure no duplicates, we looked at the possible duplicates with the age, the sex, the curusus and the year of the study: no duplicates were found. We added the sentence » No duplicates were found (checking on age, gender, cursus and year of study). »
Line 97. The authors should justify the way they measured physical activity with these questions and how they decided the categories they refer to on line 101.
We added the reference of the WHO and the article which also has chose the categories « occasional » and « regular »
Line 115. Why did the authors measure academic stress with these questions? How was the reliabiblity of the academic stress measurement assessed?
We added the reference and the sentence “the academic stress and academic satisfaction scale have sufficiently high internal consistency (Cronbach’s alphas are greater than 0.6”
Line121. The authors should justify the use of these questions regarding COVID 19 and not the use of some other validated questionnaire, since many have been developed (e.g. CAS, OCS, CRBS).
We used the same questions as in our previous studies in 2020 for external validity then we did not used developped score
Statistical analysis
The statistical analysis presented is generic. The authors should provide in further detail what models they used and under which criteria did reach inference. Were multiple comparison tests applied?
We developed the paragraph and specified the tests and the regression models
In the last paragraph of the results section (lines 169-178) there is no indication of the p- values that lead to these conclusions.
We added a sentence “The significant associated factors (p<0.05) were the following. »
Table 1. The authors should provide more information on the statistical tests that were applied and led to these conclusions. What hypotheses do the p-values refer to? Multiple comparisons should be carried out to shed light to several estimations that appear on the table.
We added in the paragraph of statistical analysis the used test
Table2. Further annotation is needed to help the reader understand what hypotheses are examined by the p-values presented.
We added in the paragraph of statistical analysis the used test and added that no ED=reference category
Discussion section.
The discussion should emphasize more on the research findings as the presentation of the literarure findings dominate the section.
In the discussion section, we have further clarified and strengthened the results of our study
All figures except figure 4 should contain more statistical information as the conclusions based on the depicted information can be mistaken.
On the graphs the p indicates a difference between the different groups, we did not make a 2 to 2 comparison
Reviewer 2 Report
Focus of the manuscript is on eating disorders during the Covid-19 pandemic. I have some comments that may strengthen the manuscript:
- Please do not use the term "influence" when using cross-sectional data. This enables the author to describe associations.
- Did the authors ask for the age of the participants. If this was not the case, it should be included in the limitation section.
- Please provide more information on the Food Security Scale.
- Please include information on validity of used instruments. Especially regarding measure of PA information are missing. If no established instrument was used for assessing PA, this should be mentioned in the limitations section.
- Please provide information on response rate.
- Please provide information on representativeness of data for the university.
- It is not clear to me, how the data are connected with Covid pandemic. We do not have data to compare how the situation was before Covid pandemic. This should be clarified in the manuscript. There might also be different trajectories during the Covid pandemic. This should be discussed, since behaviors may not be stable over time.
- Discussion, line 183: How can authors conclude that there is an increase in ED, when no data on pre-covid are available?
- Line 188: "clinical nut"??
- Please discuss social desirablility as potential limitation.
- The conclusion should focus more on the findings regarding eating disorders.
Author Response
We thank the reviewer
Focus of the manuscript is on eating disorders during the Covid-19 pandemic. I have some comments that may strengthen the manuscript:
- Please do not use the term "influence" when using cross-sectional data. This enables the author to describe associations.
The sentence has been modified “This study aimed to investigate the assoiated factors »
- Did the authors ask for the age of the participants. If this was not the case, it should be included in the limitation section.
Yes, we asked for the age, which allowed us to calculate the mean age
We added in the method section
Please provide more information on the Food Security Scale.
- Please include information on validity of used instruments. Especially regarding measure of PA information are missing. If no established instrument was used for assessing PA, this should be mentioned in the limitations section.
There was no missing data because filling in was mandatory
. We added the reference of the WHO and the article which also has chose the categories « occasional » and « regular »
We added validity information of th used instrument.
- Please provide information on response rate.
We added the participation rate 12%
- Please provide information on representativeness of data for the university.
We added the sentence in the limitation section :as a convenience sample, voluntary participation could have led to representativeness and self-selection bias as our sample had more women and healthcare students,
- It is not clear to me, how the data are connected with Covid pandemic. We do not have data to compare how the situation was before Covid pandemic. This should be clarified in the manuscript. There might also be different trajectories during the Covid pandemic. This should be discussed, since behaviors may not be stable over time.
- The same questions were asked indicating before the pandemic COVID 19 (before March 2020) and the month before
- The study encompasses the pandemic period one month before May 2021 and for eating habits, food security, and physical activity, encompasses also the pre-pandemic period: i.e. the month before the COVID-19 measures (before March 2020)”.
- Discussion, line 183: How can authors conclude that there is an increase in ED, when no data on pre-covid are available?
- Autre etude TS1C et nutrients
- Line 188: "clinical nut"??
We added the right reference [24]
- Please discuss social desirablility as potential limitation.
We added a sentence ; The study being anonymous and by self questionnaire limits the bias of desirability.
- The conclusion should focus more on the findings regarding eating disorders.
We modify the disussion
Round 2
Reviewer 1 Report
All comments have been adequately addressed.
Change "Ohter to Other" on figure 1.